# Transvenous Lead Extraction SAFeTY Score for Risk Stratification and Proper Patient Selection for Removal Procedures Using Mechanical Tools

**DOI:** 10.3390/jcm9020361

**Published:** 2020-01-28

**Authors:** Wojciech Jacheć, Anna Polewczyk, Maciej Polewczyk, Andrzej Tomasik, Andrzej Kutarski

**Affiliations:** 1Second Department of Cardiology, Medical University of Silesia in Katowice, School of Medicine with the Division of Dentistry in Zabrze, 40-055 Katowice, Poland; wjachec@interia.pl (W.J.); tomasik@onet.poczta.pl (A.T.); 2Faculty of Medicine and Health Sciences, The Jan Kochanowski University, 25-369 Kielce, Poland; Maciek.polewczyk@gmail.com; 3Department of Cardiology, Swietokrzyskie Cardiology Center, 45, Grunwaldzka St., 25-736 Kielce, Poland; 4Acute Cardiac Care Unit, Swietokrzyskie Cardiology Center, 45, Grunwaldzka St., 25-736 Kielce, Poland; 5Department of Cardiology, Medical University Lublin, 20-059 Lublin, Poland; a_kutarski@yahoo.com

**Keywords:** transvenous lead extraction, risk factors, procedural safety

## Abstract

Background: To ensure the safety and efficacy of the increasing number of transvenous lead extractions (TLEs), it is necessary to adequately assess the procedure-related risk. Methods: We analyzed potential clinical and procedural risk factors associated with 2049 TLE procedures. The TLEs were performed between 2006 and 2016 using only simple tools for lead extraction. Logistic regression analysis was used to develop a risk prediction scoring system for TLEs. Results: Multivariate analysis showed that the sum of lead dwell times, anemia, female gender, the number of procedures preceding TLE, and removal of leads implanted in patients under the age of 30 had a significant influence on the occurrence of major complications during a TLE. This information served as a basis for developing a predictive SAFeTY TLE score, where: S = sum of lead dwell times, A = anemia, Fe = female, T = treatment (previous procedures), Y = young patients, and TLE = transvenous lead extraction. In order to facilitate the use of the SAFeTY TLE Score, a simple calculator was constructed. Conclusion: The SAFeTY TLE score is easy to calculate and predicts the potential occurrence of procedure-related major complications. High-risk patients (scoring more than 10 on the SAFeTY TLE scale) must be treated at high-volume centers with surgical backup.

## 1. Introduction

Transvenous lead extraction (TLE) is considered the first-line strategy for the management of complications associated with cardiac implantable electronic devices (CIED). Recently, due to the rising incidence of CIED infections and noninfectious complications, the number of TLEs has also been increasing. According to the 2017 Report from the European Heart Rhythm Association (EHRA), more than 9000 TLEs were performed in 361 centers, making up 15 procedures per million population. The exact number of TLEs may be even higher as the available data is incomplete [1]. With the increasing incidence of difficult lead extractions, appropriate procedure-related risk assessment is becoming more and more important. According to previous studies, the rate of major complications during a TLE ranges from 0.9% to 4.0% [2,3,4,5,6,7,8,9,10,11,12], including most often damage to the heart muscle or veins. The occurrence of major complications requires immediate surgical intervention, and for this reason, extraction procedures should ideally be performed in a hybrid operating room in the presence of cardiac surgical teams. In reality, only some of the theoretically less risky procedures are performed in electrophysiological (EPS) laboratories. Therefore, procedure-related risk assessment is pivotal in terms of expanding the adoption and availability of TLE in less experienced and less equipped centers. Additionally, understanding the periprocedural risk has become imperative to properly managing CIED leads including early recognition of lead failure, proper patient selection, and providing sufficient information about the TLE, both to patients and their families.

Until now, there has been no precise percentage score for estimating the risk of developing complications during and after a TLE. The currently available TLE risk stratification tools most often consider factors that impact the technical difficulty of the procedure [7,8] or periprocedural mortality [9,10]. This study presents for the first time a validated risk score derived from a large prospective database to predict the major complications of a TLE. In addition, an easy-to-use calculator was constructed to facilitate risk assessment in clinical practice.

## 2. Methods

### 2.1. Study Population

Post hoc analysis of clinical data from 2049 patients undergoing transvenous lead extraction at a high-volume center between 2006 and 2016 was performed. The validation cohort included 551 patients operated on at the same hospital and prospectively enrolled from February 2016 to December 2017.

### 2.2. Lead Extraction Techniques

TLE was performed by a single operator, most frequently using mechanical different polypropylene Byrd dilator sheaths (Cook^®^ Medical, Leechburg, PA, USA). When removing a free-floating lead distal leaf fragment still remaining in the heart, tools for extraction via the femoral approach were used: baskets catheters (Dotter retriever basket; Cook Medical, Bloomington, IN, USA) and Amplatz Goose Neck^®^ Snare Kit (Covidien, Dublin, Ireland). To grasp a broken lead fragment, non-standard tools were used: a guide wire set dedicated for coronary sinus lead implantation, pig-tail catheters, lasso, and various hemodynamic catheters. A powered mechanical sheath system (EvolutionV R Mechanical Dilator Sheath, Cook Medical Inc., Bloomington, IN, USA; TightRail Rotating Dilator Sheath, Spectranetics, Colorado Springs, CO, USA) was used incidentally (<1.0%) as the second-line tool when the polypropylene telescoping sheaths were ineffective. Laser-assisted lead extraction was not performed in this study.

## 3. Definitions

The efficacy of the TLEs was assessed according to the 2017 Heart Rhythm Society (HRS) consensus guidelines and the ELECTRa (European Lead Extraction ConTRolled study) Registry [6,7] using the following terms:Complete procedural success: removal of all targeted leads and all lead material from the vascular space, with the absence of any permanently disabling complication or procedure-related death.Clinical success: removal of all targeted leads and all lead material from the vascular space or retention of a small portion of the lead (tip) or a small part (<4 cm) of the lead or insulation (complete or partial radiographic success), not increasing the risk of complication or perpetuation of infection or causing any permanently disabling complication or procedure-related death.

According to the ELECTRa Registry [7], intraprocedural complications were defined as any event related to the performance of the procedure that occurred or became evident from the time the patient entered the operating room or catheterization laboratory until the time the patient left the operating room. 

Failure (procedural, clinical, radiographic) referred to an inability to meet the criteria of success as appropriate [6,7].

According to the 2017 HRS consensus guidelines [6], major complications were defined as any of the outcomes related to the procedure that was life threatening, requiring significant surgical intervention, causing persistent disability, or resulting in death. In this context, procedure-related death; injury to myocardium requiring puncture, drainage, or a sternotomy; injury to large blood vessels requiring surgical repair; a pulmonary embolism requiring surgical intervention, stroke, respiratory failure, or complications associated with sedation resulting in prolonged hospitalization; and bacterial colonization of the non-infected generator pocket were considered as major complications. 

### 3.1. Data Analysis

We analyzed the number and type of leads being removed, as well as the technique of lead extraction and the effectiveness of the TLE. Univariate and multivariate logistic regression analysis was used to assess clinical and procedural factors that potentially increase the risk of developing major complications. Procedural predictors were analyzed in three categories: device-dependent, previous procedure-dependent, and directly TLE-dependent. Points were assigned to each risk factor based on the odds ratio (OR). The total number of points served as a basis for developing a risk scoring system forming the acronym SAFeTY TLE to predict periprocedural complications. To calculate the risk score, points were assigned to each risk factor based on the odds ratio. Patients were divided into groups depending on the sum of points. In each subgroup, the number of major complications or no complication status was recorded. The data were then used to develop a risk curve. The logistic function was used to determine the relationship between the number of points and the probability of major complications. 

The validation of the SAFeTY TLE score was based on a prospective analysis of data obtained from 551 consecutive patients undergoing a TLE at the same hospital from February 2016 to December 2017.

All patients signed an informed consent form to undergo the proposed procedure and participate in the study. Clinical endpoints were established for all patients. The exact date of death was obtained from the patient’s medical records, relatives, or a national identity database.

The study was approved by the local Bioethics Committee and the ethic approval Code was 26/2017.

### 3.2. Statistical Analysis 

The patients were divided into two groups according to the occurrence of major complications. The Shapiro–Wilk test was used to test the normality of the continuous data, both for the whole group and the subgroups. Continuous data were presented as means and standard deviations. The follow-up period after a TLE is expressed as the median and interquartile range (IQR). Categorical data are presented as numbers and percentages. Due to disproportions resulting from the small size of the group with major complications, the significance of baseline inter-group differences was determined using the χ^2^ test with a Yates correction or the unpaired “U” Mann–Whitney test, as appropriate. Kaplan-Meier survival analysis and the log-rank test were used to assess the overall survival after a TLE.

In all groups, logistic regression analysis (logit regression) was performed to examine the relationship between variables and the occurrence of major complications. Variables with *p*-values < 0.10 in univariate analysis were entered into the multivariate logistic regression model. The regression results were reported as odds ratios with the corresponding 95% confidence intervals (CIs). The receiver operating characteristic (ROC) curve analysis was performed to determine the optimal cut-off value of the sum of lead dwell times, hemoglobin concentration, and a score above which the risk of major complications significantly increased. The Horsmer–Lemeshow test was used to assess whether the observed event rate in the derivation group matched the expected event rate in the validation group. Additionally, the sensitivity, specificity, and likelihood ratios were calculated for the validation cohort. Results yielding a *p*-value less than 0.05 were considered significant. Statistical analysis was performed using the STATISTICA 10.0 software (StatSoft Inc., Tulsa, OK, USA).

## 4. Results

In the present study, a total of 3425 leads were removed from 2049 patients. The most common approach was lead removal using Byrd dilators via the subclavian vein (1758 procedures (85.8%)). Other techniques included simple traction (screw-out and traction, 360 leads (10.5%)), a combined approach (superior and inferior, 65 leads (1.9%)), femoral lead extraction using various tools (45 leads (1.3%)), and a jugular approach using lasso catheters and Byrd dilators (4 leads (0.1%)). The study also included nine leads (0.26%) liberated using transvenous extraction from patients initially referred for open heart surgery because of the presence of large vegetations or coexisting valvular pathology. 

Complete procedural success and clinical success was achieved in 95.0% and 97.9% of patients, respectively. Major complications occurred in 37 (1.81%) patients, including periprocedural death in 8 (0.39%) patients. 

In the derivation cohort, the follow-up lasted up to 10 years. The mean follow-up time was 3.37 (±2.29) years, with the median follow-up being 3.17 years (IQR: 1.41–5.03). A total of 459 (22.40%) patients died within this time period. In the validation cohort, the follow-up lasted up to 2 years. The mean follow-up time was 1.07 (±0.50) years, with the median follow-up being 0.93 years (IQR: 0.47–1.44). A total of 44 (7.99%) patients died in the same timeframe. The Kaplan–Meier survival curves did not differ significantly between the groups. The results are presented in Figure 1.

Table 1 summarizes the baseline characteristics of patients divided according to the occurrence of major complications during a TLE and the outcome of the univariate logistic regression. Univariate regression analysis demonstrated that the occurrence of major complications positively correlated with the following variables: lead dwell time (most adequately the sum of lead dwell times), female gender, the severity of anemia, the number of prior CIED-related procedures, and the number and type of extracted leads. The above variables were entered into a multivariate linear regression model (Table 2).

A simple scoring system was developed to predict the risk associated with transvenous lead extraction based on the variables with the level of significance below 0.1. Table 3 summarizes the results of multivariate regression analysis and the number of points assigned for each variable.

As risk stratification and patient selection for a TLE may be clinically challenging, a scoring system was developed and represented using the acronym SAFeTY TLE where:
Ssum of lead dwell times6.095 pointsAanemia2.291 pointsFefemale gender2.740 pointsTtreatment (previous procedures)1.364 points for each procedureYyoung patient (first implantation under the age of 30)2.174 pointsTLEtransvenous lead extraction

The sum of the risk points correlated with the probability of developing major complications during a TLE and the relationship was expressed as the logistic function in the following equation:risk of major complications (%) = 100/(1 + 644/(1.3213^x^)),
where “x” is the number of points obtained (Figure 2) 

Based on this formula a simplified calculator was created to predict the risk of major complications during the TLE procedure (Figure 3, calculator is online at http://alamay2.linuxpl.info/kalkulator/).

A score of 9.933 (Figure 4A) was considered the clinical cut-off to indicate patients at a higher procedural risk. A SAFeTY TLE score ≥ 16 implied a very high risk of complications, exceeding 11.82%. In the validation cohort, there were seven major complications (1.27%). The results of the Hosmer–Lemeshow test (*p* = 0.424) showed that the expected event rates in the validation group matched the observed event rates in the derivation group. In the validation group, a receiver operating characteristic (ROC) curve analysis demonstrated that 9.933 points on the TLE SAFeTY scale corresponded to at least a 2.410% probability of major complications with a sensitivity of 85.714% and specificity of 76.839% (AUC = 0.856, 95% CI (0.747–0.966), *p* < 0.000) (Figure 4B).

The list of major complications in the study population along with the SAFeTY TLE risk assessment was presented in Table 4.

In the validation cohort, there were seven major complications (1.27%). The results of the Hosmer–Lemeshow test (*p* = 0.424) showed that the expected event rates in the validation group matched the observed event rates in the derivation group. In the validation group, a receiver operating characteristic (ROC) curve analysis demonstrated that 9.933 points on the SAFeTY TLE scale corresponded to at least a 2.410% probability of major complications with a sensitivity of 85.714% and specificity of 76.839% (AUC = 0.856, 95% CI (0.747–0.966), *p* < 0.000) (Figure 4B). The type of major complication with SAFeTY TLE points in the validation cohort is presented in Table 5.

## 5. Discussion

Transvenous lead extraction has only been known for a relatively short period of time as it was introduced into clinical practice in the 90s [13,14]. For this reason, risk assessment of the procedure can still be difficult. Available scoring systems are based on a relatively small number of cases and incorporate various potential risk factors [1,7,8,9,10,15,16]. Additionally, the early scales focused mainly on technical difficulties and factors influencing short-term mortality [7,8,9,10,11,12]. Only two existing scores estimate the risk of major complications during TLE. One of them, based on 702 procedures, differentiates high-risk patients (pacemaker (PM) lead dwell time >10 years or implantable cardioverter-defibrillator (ICD) lead dwell time >5 years), moderate-risk (PM lead dwell time 1–10 years or ICD lead dwell time 1–5 years), and low-risk patients (any lead dwell time <1 year) with major complication rates of 5.3%, 1.2%, and 0%, respectively (*p* < 0.001 [15]). The second score, based on data from 187 subjects, identifies patients at intermediate and high-risk for TLE complications, taking into account lead dwell time (PM lead >10 years and ICD lead >5 years) and the coexistence of congenital heart disease, age <15 at time of implantation, hemodialysis, calcified superior vena cava, or myocardium adjacent to the lead on a chest radiography or CT scan, active sepsis, heart failure, and New York Heart Association (NYHA) functional class IV [16]. The current SAFeTY TLE score is based on a much larger population of 2049 patients and includes a more extensive assessment of the potential risk of major complications. It is worth noting that the proposed score includes a complex factor that evaluates the impact of the lead dwell time on the risk of complications. The sum of lead dwell times was the most sensitive parameter, which is a combination of age and the number of leads. In practice, the risk associated with extracting a larger number of leads with shorter dwell times is equivalent to the risk of extracting an older lead (e.g., removal of three leads implanted for 6 years may be as dangerous as the extraction of one lead implanted for 18 years).

The previous risk stratification tools assess the technical difficulty of lead removal depending on a given parameter. Based on data from 208 patients, a 4-stage scale was developed for predicting the difficulty of advanced extraction techniques. The following predictors were identified: age of patients <70.7 years, implant duration >37 months, and the extraction of at least two leads with one of them being a defibrillator lead [7]. Other investigators proposed the Lead Extraction Difficulty (LED) index for technical risk assessment during TLE by considering several factors that affected the fluoroscopy time as an index of difficulty. The LED score was calculated as the number of extracted leads within a procedure + lead age (years from implant), +1 if dual-coil/−1 if vegetation on the lead; a value greater than 10 predicted a fluoroscopy time above the 90th percentile [8]. A direct comparison of the factors influencing the technical difficulties during TLE and the occurrence of major complications is not possible because procedural safety is not entirely dependent on technical issues. Multivariate analysis, used to construct the SAFeTY TLE score, demonstrated that anemia, female sex, previous procedures, and a younger age were stronger predictors of major complications rather than the technical problems themselves.

Other risk scores focus on short-term mortality after a TLE. A risk nomogram for predicting 30-day, all-cause mortality was created using data from 2999 TLE procedures and included the following factors: age, body mass index (BMI), hemoglobin, end stage renal disease, ejection fraction, NYHA functional class, lead extraction due to infection, prior TLE, and extraction of a dual-coil lead. This scale was characterized by a high concordance index value of 0.867 [9]. Another scoring system, known by its acronym IKAR (infective indications [1 point], kidney dysfunction [2 points], age >56 [1 point], removal of high voltage lead [1 point]), was based on data from 130 patients and predicted one-year mortality after TLE. Mortality stratified by the IKAR score was as follows: 0 points, 0%; 1–2 points, 16%; ≥3 points, 79%, and ≥4 points, 94% [10]. 

The SAFeTY TLE score not only combines the conventional factors and those discussed above, but it also takes into account a wider impact of risk factors on procedural safety. Similar risk factor information has been collected for meta-analysis and large registries. According to the National Cardiovascular Registry, which includes 762 TLE centers, predictors of major perioperative complications were: female sex, admission other than electively for procedure, ≥3 leads extracted, longer implant duration, dislodgement of other leads, and patient’s clinical status requiring lead extraction (infection/perforation) [17]. Analysis of the European Lead Extraction ConTRolled (ELECTRa) Registry demonstrated that the risk factors of cardiac avulsion or tear with tamponade were: extraction of a Riata lead, female sex, mean lead dwelling time of more than 10 years, extraction of 3 or more leads, or placement of multiple sheaths. Additionally, occlusion or critical stenosis of superior venous access and mean lead dwelling time of more than 10 years were independent predictors of vascular avulsion or tearing [18].

By comparing the available risk stratification tools, the present study shows that each scoring system is intended for a different purpose, and there is no uniform, simple to use percentage scale for predicting TLE-related risk. There is a significant need for precise risk assessment to ensure procedural safety. The proposed SAFeTY TLE scale should facilitate the proper and risk-adjusted performance of the procedure by taking into account the venue of lead extraction and the presence of the most experienced first operator and scrubbed cardiac surgeon. It is currently recommended to perform the procedure in a hybrid operating room or cardiac surgery operating theater equipped with a good-quality X-ray machine; however, such operating rooms may not always be available. Certainly, the SAFeTY TLE score facilitates identification of high-risk patients (a score of 10.0 denotes at least a 2.46% risk of major complications), in whom TLE should not be performed without appropriate supervision, even in an electrophysiology lab. On the other hand, patients at low risk of major complications (0.47%) could be considered for TLE in the setting of an electrophysiology lab. The easy-to-use SAFeTY TLE calculator available online (http://alamay2.linuxpl.info/kalkulator/) should become a helpful adjunct in clinical decision-making regarding lead extraction, especially in less experienced centers. Detailed quantitative risk assessment offers the opportunity to refer high-risk patients to high-volume hospitals. 

## 6. Limitations

The limitation of the present study is the operator’s use of only manual extraction with femoral tools, although this should encourage the validation of the scale on populations undergoing lead extraction using the other techniques. Furthermore, it would be highly desirable to validate the SAFeTY TLE score on the ELECTRa population.

## 7. Conclusions

Stratification of TLE-associated risk is extremely important to ensure a high level of procedural effectiveness and safety; therefore, a simple multiparametric algorithm is required to facilitate the prediction of potential significant complications. In order to improve procedural effectiveness, a novel user-friendly SAFeTY TLE score was developed on the basis of clinical data obtained from a large population of patients undergoing transvenous lead extraction procedures.

## Figures and Tables

**Figure 1 jcm-09-00361-f001:**
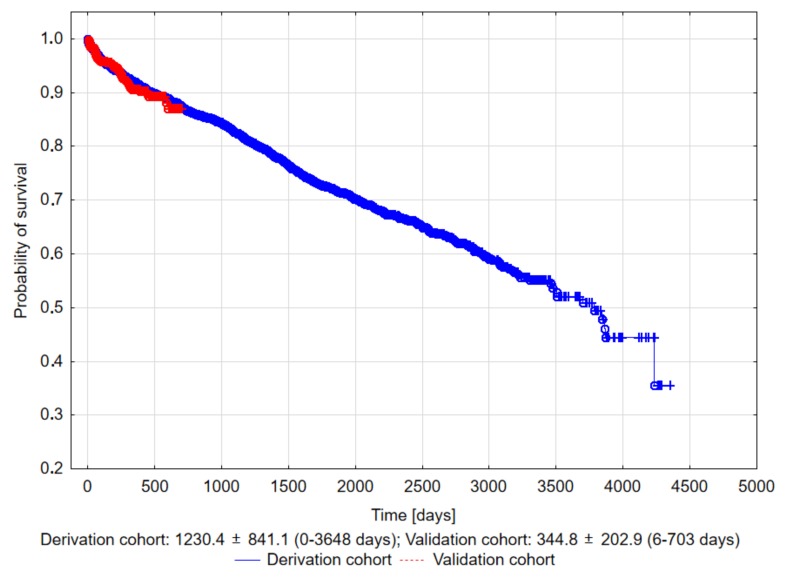
The Kaplan–Meier curve showing the probability of survival after a transvenous lead extraction (TLE) in the derivation and validation cohorts, *p* = 0.748.

**Figure 2 jcm-09-00361-f002:**
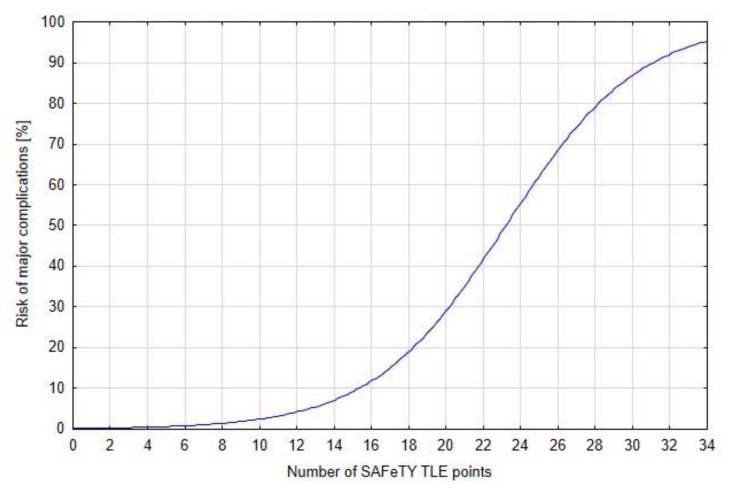
SAFeTY TLE algorithm—dependency between the risk score and the risk of major complications, results for 2049 TLE procedures. Logistic equation for the occurrence of major complications (%) = 100/(1 + 644/(1.3213^x^).

**Figure 3 jcm-09-00361-f003:**
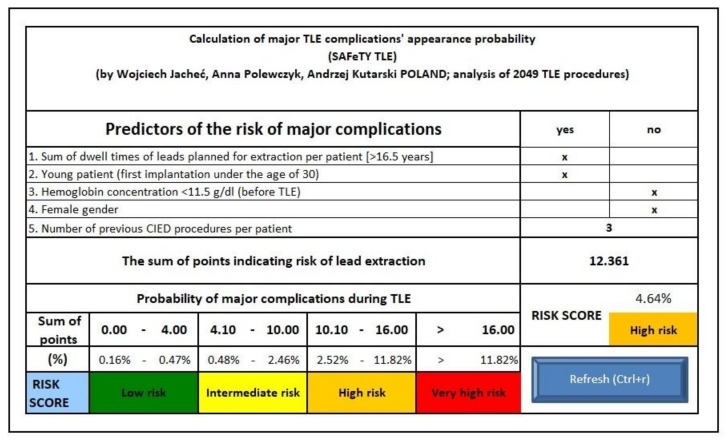
Risk calculator to predict the risk of major complications during a transvenous lead extraction using a SAFeTY TLE score.

**Figure 4 jcm-09-00361-f004:**
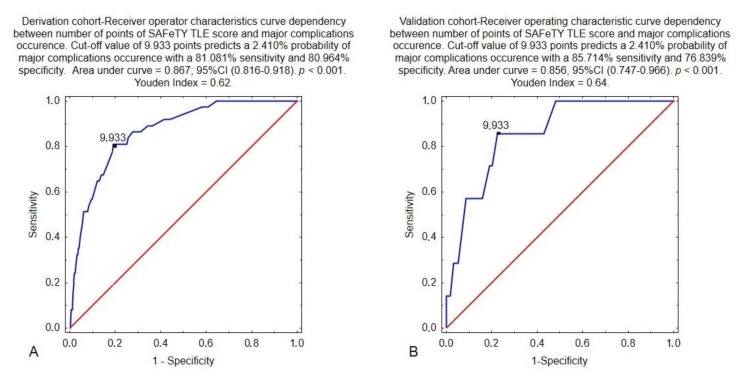
Derivation cohort—Receiver Operating Characteristic curve showing dependency between the number of SAFeTY TLE points and the occurrence of major complications (**A**). Validation cohort—Receiver Operating Characteristic curve showing dependency between the number of SAFeTY TLE points and the occurrence of major complications (**B**).

**Table 1 jcm-09-00361-t001:** Baseline characteristics of the study group depending on the occurrence of major complications during a TLE, along with a univariate logistic regression.

	Major Complications	Without Major Complications	χ^2^/U Mann-Withney	Univariate Logistic Regression		
*n* = 2049	37	2012	*p*	OR	95% CI	*p*
Age in the time of TLE	64.03 ± 16.62	64.99 ± 15.84	0.750	0.996	0.977–1.016	0.713
Age < 30 year, *n* (%)	8 (21.622)	167 (8.302)	0.010	3.041	1.368–6.762	0.006
Male sex, *n* (%)	12 (32.432)	1237 (61.481)	0.000	0.306	0.153–0.613	0.001
Female sex, *n* (%)	25 (67.568)	775 (38.519)	0.000	3.270	1.633–6.549	0.001
NYHA functional class I–II vs. III–IV, *n* (%)	33/4 (89.19/10.81)	1757/255 (87.33/12.67)	0.930	0.718	0.435–1.182	0.192
LVEF (change by 10%)	45.556 ± 8.433	41.541 ± 10.950	0.032	1.530	1.032–2.270	0.034
Creatinine concentration >2mg %, *n* (%)	4 (10.81)	108 (5.37)	0.281	2.190	0.760–6.308	0.146
Hemoglobin concentration (g/dL)	11.975 ± 2.174	13.110 ± 1.883	0.000	0.760	0.650–0.882	0.001
Anticoagulant therapy, *n* (%)	9 (24.32)	690 (34.29)	0.275	0.613	0.288–1.306	0.204
Antiplatelet therapy, *n* (%)	11 (29.73)	898 (44.63)	0.101	0.560	0.275–1.140	0.110
Infective indications, *n* (%)	14 (37.8)	801 (39.8)	0.941	0.821	0.416–1.620	0.569
Pocket infection, *n* (%)	5 (13.51)	580 (28.82)	0.063	1.179	0.860–1.601	0.291
Lead related infective endocarditis, *n* (%)	12 (32.43)	537 (26.69)	0.552	1.168	0.573–1.238	0.669
Number of procedures before TLE	3.216 ± 1.766	1.823 ± 1.104	0.000	1.785	1.502–2.121	0.000
Leads on the either side of chest, *n* (%)	6 (16.216)	85 (4.225)	0.000	4.335	1.761–10.67	0.001
TLE for leads implanted on both sides in the chest wall, during the same TLE procedure, *n* (%)	5 (13.514)	30 (1.491)	0.000	10.32	3.759–28.32	0.000
Age of the oldest extracted lead (years)	19.145 ± 19.014	7.482 ± 5.686	0.000	1.162	1.118–1.207	0.000
Mean age of extracted lead (years)	16.523 ± 18.574	6.791 ± 4.885	0.000	1.194	1.138–1.253	0.000
Sum of lead dwell times planned for extraction (years)	28.532 ± 20.229	11.718 ± 10.665	0.000	1.072	1.053–1.092	0.000
Number of abandoned leads in patients eligible for extraction	0.705 ± 1.052	0.187 ± 0.518	0.000	2.400	1.720–3.351	0.000
Presence of abandoned lead(s) before TLE, *n* (%)	14 (37.838)	269 (13.370)	0.000	3.944	2.004–0.762	0.000
High probability of scar tissue binding the leads, *n* (%)	7 (19.919)	101 (5.020)	0.000	3.624	1.478–8.888	0.005
TLE of ICD lead, *n* (%)	4 (10.811)	515 (25.60)	0.063	0.345	0.122–0.979	0.045
Targeted extraction of RA lead (any), *n* (%)	30 (81.081)	1169 (58.101)	0.008	3.091	1.350–7.075	0.008
Planned extraction of UP A lead, *n* (%)	7 (18.919)	96 (4.771)	0.000	4.706	2.014–10.99	0.000
Considered extraction of UP V lead, *n* (%)	8 (21.622)	188 (9.344)	0.026	2.675	1.205–5.938	0.016
TLE of UP leads, *n* (%)	10 (27.027)	231 (11.481)	0.008	2.854	1.363–5.974	0.005
TLE of UP leads above the median of age, *n* (%)	11 (29.730)	157 (7.803)	0.000	1.739	1.121–2.698	0.014
TLE of inactive leads, *n* (%)	18 (48.649)	320 (15.905)	0.000	2.117	1.427–3.141	0.000
TLE leads on either side of chest, *n* (%)	5 (13.514)	30 (1.491)	0.000	10.32	3.759–28.319	0.000
TLE of the more than three leads, *n* (%)	5 (13.514)	76 (3.777)	0.009	3.978	1.507–10.50	0.005
Disruption of lead during TLE, *n* (%)	5 (13.514)	71 (3.529)	0.006	4.272	1.615–11.30	0.003
Any technical problem, *n* (%)	14 (37.838)	311 (15.457)	0.000	3.329	1.694–6.542	0.001
Number of big technical problems, *n* (%)	0.432 ± 0.689	0.142 ± 0.413	0.000	2.464	1.559–3.894	0.001

Abbreviations: ICD—implantable cardioverter-defibrillator; LVEF—left ventricular ejection fraction; NYHA—New York Heart Association; RA—right atrium; TLE—transvenous leads extraction; UP—unipolar; UP A—unipolar atrial; UP V—unipolar ventricular.

**Table 2 jcm-09-00361-t002:** Multivariate analysis of major complications.

Major Complications	Multivariate Regression		
	OR	95% CI	*p*
Sum of dwell times of leads planned for extraction per patient >16.5 years	6.095	2.299–16.16	0.000
Young patient (first implantation under the age of 30)	2.174	0.881–5.368	0.092
Hemoglobin concentration <11.5 g/dL	2.291	1.127–4.655	0.022
Female gender	2.740	1.310–5.732	0.007
LVEF (change by 10%)	0.827	0.605–1.129	0.231
Number of procedures before TLE	1.364	1.048–1.774	0.021
Leads on the either side of chest	0.388	0.045–3.371	0.391
Need for extracting leads implanted on both sides in the chest wall during the same TLE procedure	1.250	0.445–3.507	0.672
High probability of scar tissue binding the leads-	1.125	0.577–2.192	0.730
TLE of ICD lead	0.576	0.174 ± 1,192	0.367
Number of abandoned leads in patients eligible for extraction	5.055	0.485–52.63	0.175
Planned extraction of four or more leads during TLE procedure	0.697	0.185–2.629	0.594
Presence of abandoned lead(s) before TLE	1.099	0.338–3.576	0.875
Targeted extraction of RA lead (any)	0.954	0.339–2.680	0.928
Planned extraction of UP A lead	1.538	0.527–4.490	0.431
Considered extraction of UP V lead	0.495	0.170–1.441	0.197

Abbreviations: ICD—implantable cardioverter-defibrillator; LVEF—left ventricular ejection fraction; RA—right atrium; TLE—transvenous leads extraction; UP A—unipolar atrial; UP V—unipolar ventricular.

**Table 3 jcm-09-00361-t003:** Number of points predicting the risk associated with TLE as a result of the multivariate analysis of major complications.

Major Complication	OR (SAFeTY-TLE Points)	95% CI	*p*
Sum of the dwell times of leads planned for extraction per patient (>16.5 years)	6.095	2.299–16.12	0.000
Young patient (first implantation under the age of 30)	2.174	0.881–5.368	0.092
Hemoglobin concentration <11.5 g/dL (before TLE)	2.291	1.127–4.655	0.022
Female gender	2.740	1.310–5.732	0.007
Number of previous CIED procedures per patient (for each procedure)	1.364	1.048–1.774	0.021
**Risk Assessment Using TLE SAFeTY Score**			
**TLE SAFeTY**	**Risk Score**		
0.00–4.00	Low risk (0.16–0.47%)		
4.10–10.00	Moderate risk (0.48–2.46%)		
10.1–16.00	High risk (2.52–11.82%)		
>16.00	Very high risk (>11.82%)		

Abbreviations: CIED—cardiac implantable electronic devices; TLE—transvenous leads extraction.

**Table 4 jcm-09-00361-t004:** List of major complications with an assessment of the number of SAFeTY TLE points in the study population.

Major Complications: Appearance, Kind of Injury, Rescue Procedures, and Outcome	No (%) of MJC	% of All TLE	Place of Structural Damage					No of SAFeTY TLE Points	Fatal Outcome
			RA	RV	CS	SVC	Other		*n* (%)
Hemopericardium—cardiac surgery	19 (51.35)	0.93%	13	3	3	0	0	12.05 ** (3.66–19.18) ***	6 (31.6)
Hemopericardium—pericardiocentesis—effective drainage	11 (29.73)	0.54%	10*	2*	0	0	0	14.72 ** (10.20–18.76) ***	0 (0.0)
Hemothorax—pleural drainage	3 (8.11)	0.15%	0	0	0	3	0	11.07 ** (4.10–15.22) ***	0 (0.0)
Hemothorax—thoracic surgery	1 (2.70)	0.05%	0	0	0	1	0	7.769	0 (0.0)
Brain emboli—stroke—rehabilitation	1 (2.70)	0.05%	0	0	0	0	1	10.19	0 (0.0)
Gradual decrease of contractility and delayed death (no structural damage)	1 (2.70)	0.05%	0	0	0	0	1	5.47	1 (100)
Pulmonary embolism—cardiac surgery	1 (2.70)	0.05%	0	0	0	0	1	13.85	1 (100)
All patients with major complications	37 (100)	1.81%	23 *	5 *	3	4	3	12.54 ** (3.55–19.18) ***	8 (21.62)
All patients without major complications	2012 (0.00)	98.19%	0	0	0	0	0	5.66 **^,AAA^ (1.36–22.85) ***	0 (0.00)

*—one case of both perforation of right atrium and ventricle, **—mean, ***—min-max. ^AAA^—*p* < 0.001 when compared to patients with major complications. Abbreviations: CS—coronary sinus; MJC—major complications; RA—right atrium, RV—right ventricle, SVC—superior vena cava; TLE—transvenous leads extraction.

**Table 5 jcm-09-00361-t005:** Major complications with SAFeTY TLE points in the validation cohort.

Major Complications: Appearance, Kind of Injury, Rescue Procedures, and Outcome in Validated Group	No (%) of MJC	% of All TLE	Place of Structural Damage					No of SAFeTY TLE Points	Fatal Outcome
			RA	RV	CS	VCS	Other		*n* (%)
Hemopericardium *—cardiac surgery	7 (100.0)	1.27%	4	3	0	0	0	13.82 ** (6.40–24.77) ***	0 (0.00)
All patients without major complications	544 (0.00)	98.73%	0	0	0	0	0	6.65 **^,AA^ (1.36–17.95) ***	0 (0.00)

*—one case of both hemopericardium and hydrothorax, **—mean, ***—min-max. ^AA^—*p* < 0.01 when compared to patients with major complications. Abbreviations: MJC—major complications; TLE—transvenous leads extraction.

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
