# Peer review of "Transvenous Lead Extraction SAFeTY Score for Risk Stratification and Proper Patient Selection for Removal Procedures Using Mechanical Tools"

_jcm, 2020, doi:10.3390/jcm9020361_

Round 1
Reviewer 1 Report
Reviewer comment:
In this single-center study, Jachec et al. analyzed potential clinical and procedural risk factors associated with transvenous lead extractions (TLE) procedures and develop a risk predictive scoring syste, TLE SAFeTY Score, which was validated in the prospective patients’ cohort. Although this study contains important clinical implications, there exists several major concerns.
Major concerns:
#1. As the authors stated in the method section, major complications included procedure-related death, injury to myocardium requiring puncture, drainage or sternotomy, injury to large blood vessels requiring surgical repair, pulmonary embolism requiring surgical intervention, stroke, respiratory failure or complications associated with sedation resulting in prolonged hospitalization, and bacterial colonization of the non-infected generator pocket. The authors assessed the risk factors for overall major complications and concluded that lead dwell time (most adequately the sum of lead dwell times), female gender, the severity of anemia, the number of prior CIED-related procedures and the number and type of extracted leads were independent predictors. However, the risk factors for cardiac avulsion or vascular tear might be different from those for other complications such as stroke, respiratory failure, complications associated with sedation. The authors should examine the breakdown of the major complications to determine the individualized risk factors for cardiac avulsion or vascular tear, and others.
#2. The authors performed multivariate analysis by using the variables associated with the occurrence of major complications in the univariate analysis. Lack of other factors in the univariate analysis shown in table 1 including as age of the oldest lead, any technical problem, and number of the big technical problems may have had significant impact on the results of multivariate analysis. Furthermore, in the multivariate analysis. sum of dwell times of leads planned for extraction per patients > 16.5 was not the factor assessed in the univariate analysis, and young patients (first implantation under the age of 30) may also be an associated variable with the sum of the dwell times.
#3. As stated in the limitation section, the studied population included only manual extraction with femoral tools. Therefore, the title need to be revised as ‘Transvenous Lead Extraction SAFeTY Score for Risk Stratification and Proper Patient Selection for “Manual” Removal Procedures.
Author Response
Response to Reviewer 1
We appreciate Reviewer’s comments on our manuscript. Please find below answers to the specific comments:
Reviewer 1.
Comment 1 As the authors stated in the method section, major complications included procedure-related death, injury to myocardium requiring puncture, drainage or sternotomy, injury to large blood vessels requiring surgical repair, pulmonary embolism requiring surgical intervention, stroke, respiratory failure or complications associated with sedation resulting in prolonged hospitalization, and bacterial colonization of the non-infected generator pocket. The authors assessed the risk factors for overall major complications and concluded that lead dwell time (most adequately the sum of lead dwell times), female gender, the severity of anemia, the number of prior CIED-related procedures and the number and type of extracted leads were independent predictors. However, the risk factors for cardiac avulsion or vascular tear might be different from those for other complications such as stroke, respiratory failure, complications associated with sedation. The authors should examine the breakdown of the major complications to determine the individualized risk factors for cardiac avulsion or vascular tear, and others.
Response 1 Yes, the methodology section presents a list of major complications in accordance with HRS 2017 guidelines, however, the current material was dominated by complications related to cardiac and vascular damage. We present analyzed major complications in additional tables (Table IV and Table V). The most of them (study group and validation cohort) were associated with mechanical effects on the cardiovascular system. In presented group there were no complications associated with anesthesia, infectious complications or respiratory failure. Only one case – ischaemic stroke was probably connected with embolic mechanism. Therefore, the factors assessed are risk factors for cardiac avulsion and vascular tear.
Comment 2 The authors performed multivariate analysis by using the variables associated with the occurrence of major complications in the univariate analysis. Lack of other factors in the univariate analysis shown in table 1 including as age of the oldest lead, any technical problem, and number of the big technical problems may have had significant impact on the results of multivariate analysis. Furthermore, in the multivariate analysis. sum of dwell times of leads planned for extraction per patients > 16.5 was not the factor assessed in the univariate analysis, and young patients (first implantation under the age of 30) may also be an associated variable with the sum of the dwell times.
Response 2 Variable „age of the oldest lead” was not used in univariate analysis, because among 3 parameters assessed age of the extracted leads- the most significant was „sum of lead dwell time”:
|
|
OR |
95%CI |
p |
|
Age of the oldest extracted lead |
1,070 |
0,964-1,187 |
0,203 |
|
Mean age of extracted lead |
1,046 |
0,930-1,176 |
0,452 |
|
Sum of lead dwell times planned for extraction |
1,034 |
1,004-1,065 |
0,027 |
The idea of calculator is to assess the risk of patients qualified for TLE in terms of selecting a group at particularly high risk, based on the baseline data available at the stage of procedures planning (techniques, sites, security). Therefore, the technical problems that appeared during TLE were not included in the analysis.
Sum of dwell times of leads planned for extraction per patients was presented in the table with univariate analysis, although it has not been precisely described (it appears as "Sum of lead dwell times" instead of "Sum of lead dwell times planned for extraction" – it was corrected). There is indeed a difference between the lead dwell time in young and older people: sum of leads dwell time ≤30 y.o. 17,76±10,57 vs >30 y.o 10,57±14,53 P<0,001, but we did not find any statistically significant interaction between the age of the patients (≤30 years and> 30 years) and the sum of leads dwell time:
|
Major complication |
OR |
95%CI |
p |
|
Sum of dwell times of leads planned for extraction per patient [>16.5 years] and young patient (first implantation under the age of 30) |
1,075 |
0,120-9,667 |
0,948 |
Comment 3 As stated in the limitation section, the studied population included only manual extraction with femoral tools. Therefore, the title need to be revised as ‘Transvenous Lead Extraction SAFeTY Score for Risk Stratification and Proper Patient Selection for “Manual” Removal Procedures.
According to the methodology, the study population included patients whose basic technique for leads extraction was the use of mechanical tools. Only laser energy was not used. We have introduced precise corrections to the „Methods” section. The title of manuscript has been changed to „Transvenous Lead Extraction SAFeTY Score for Risk stratification and proper patient selection for removal procedures using mechanical tools”.
Reviewer 2 Report
The authors provide an interesting work in the field of transvenous lead extractions. Based on an extensive number of patients, they have devised and validated a very practical score to assess the risk of major complications for patients which could help decide which patients can be extracted with or without surgical backup.
Limitations include that all procedures were performed by a sole operator.
One wonders whether a higher use of powered extraction tools such as Evolution, TightRail or Laser would have influenced the rate of complications as their use is not rare in big registries.
Minor remarks:
add abbreviations in the table legends the authors did not mention in what proportion of patients a locking stylet was used references according to JCM's style and not by alphabetical order
Author Response
Reviewer 2
Thank you very much for your valuable comments. We have made every effort to improve the manuscript according to your suggestion.
The authors provide an interesting work in the field of transvenous lead extractions. Based on an extensive number of patients, they have devised and validated a very practical score to assess the risk of major complications for patients which could help decide which patients can be extracted with or without surgical backup.
Comment 1 Limitations include that all procedures were performed by a sole operator.
Response 1 All procedures were performed by one first operator; but in most cases he worked side by side with several experienced co-operators, frequently well-trained cardiac surgeon
Comment 2 One wonders whether a higher use of powered extraction tools such as Evolution, TightRail or Laser would have influenced the rate of complications as their use is not rare in big registries.
Response 2 From scientific point of view, only prospective studies Evelution/TighRail as the first line tool vs Evolution/TighRail as the second option (rescue) tool can answer for this question. If Evolution are utilized as the rescue (but more dangerous and ten time more expensiwe) option – it means, that this tools were utilized in very most difficult cases and most terrible procedures. And major complications have to be much more frequent in such situations.
Last author present and hot calculation contains the comparison TLE major complication and necessity to perform urgent rescue cardiac surgery when Evelution/TighRail was utilized and not utilized. The answer for Reviewer question is yes, but only unknown part of major complications seems to be Evolution related (damage of SVC seems to be most typical, but due to very rare appearance the statistic is impossible).
|
Most recent A Kutarski’ last author) experience |
NO |
YES |
All patients |
|||
|
Evolution, TighRail |
Yes |
% |
No |
% |
No |
% |
|
Number of procedures |
3144 |
100,0% |
41 |
100,0% |
3185 |
100,0% |
|
Major complictions - All |
55 |
1,7% |
6 |
14,6% |
61 |
1,9% |
|
Rescue cardiac surgery |
34 |
1,1% |
4 |
9,7% |
38 |
1,2% |
Minor remarks:
Comment 1 Add abbreviations in the table legends
Response 1 The abbreviations have been added to the table legends
Comment 2 The authors did not mention in what proportion of patients a locking stylet was used
Response 2 The locking stylets – (the Liberator Locking Stylet - Cook Medical, Bloomington, Indiana, USA and the Lead Locking Device (LLD) Spectranetix, Colorado Springs,) were utilized habitually only during extraction leads having dwell time more than twenty years, unipolar atrial and right ventricular leads and old models of passive leads, when increased risk of lead break was suspected. For extraction of all models of screw-in leads and other passive isodiametric leads lucking stylets were not utilized due to economic reasons. The standard stylets were put-upon in predominant percentage of procedures.
Comment 3 References according to JCM's style and not by alphabetical order
Response 3 References have been corrected
Round 2
Reviewer 1 Report
The authors revised the manusctipt properly according to the reviewer's comments.